# Cardiovascular Risk Factors and Their Association with Vitamin D Deficiency in Mexican Women of Reproductive Age

**DOI:** 10.3390/nu11061211

**Published:** 2019-05-28

**Authors:** Alejandra Contreras-Manzano, Salvador Villalpando, Claudia García-Díaz, Mario Flores-Aldana

**Affiliations:** Center for Nutrition and Health Research, National Institute of Public Health, Cuernavaca 62100, Mexico; alejandra.contreras@insp.mx (A.C.-M.); clauw.g.diaz@gmail.com (C.G.-D.); mario.flores@insp.mx (M.F.-A.)

**Keywords:** vitamin D deficiency, 25-OH-D, women, cardiovascular risk factors, T2DM, obesity

## Abstract

Based on a nationally representative sample of young Mexican women aged 20 to 49 years (*n* = 3260), we sought to explore whether cardiovascular risk factors and acute myocardial infarction (AMI) were associated with vitamin D deficiency (VDD, defined as 25-OH-D <50 nmol/L). To this end, we obtained sociodemographic, serum and anthropometric data from the 2012 National Health and Nutrition Survey (ENSANUT 2012). Analyses were developed through logistic regression models adjusted for potential confounders. The prevalence of VDD was significantly higher in obese women (42.5%, 95% CI; 37.3–47.9) compared to women with a normal body mass index (29.9%, 95% CI; 23.5–37.1, *p* = 0.05), in those with high total cholesterol (TC) (45.6% 95% CI; 39.4–51.9) compared to those with normal TC levels (33.9%, 95% CI 30–38.1, *p* = 0.03), and in those with insulin resistance (IR) (44%, 95% CI; 36.9–51.7) or type 2 diabetes mellitus (T2DM) (58.6%, 95% CI 46.9–69.4) compared to those with normal glycemia (no insulin resistance: 34.7%, 95% CI; 30.9–38.8, *p* = 0.04 and no T2DM: 34.9%, 95% CI 31.4–38.6, *p* < 0.001). Utilizing individual models to estimate cardiovascular risk according to VDD, we found that the odds of being obese (odds ratio, OR: 1.53, 95% CI 1.02–2.32, *p* = 0.05), or having high TC levels (OR: 1.43, 95% CI; 1.05–2.01, *p* = 0.03), T2DM (OR: 2.64, 95% CI; 1.65–4.03, *p* < 0.001), or IR (OR: 1.48, 95% CI 1.04–2.10, *p* = 0.026) were significantly higher in women with VDD (*p* < 0.05). Odds were not statistically significant for overweight, high blood pressure, sedentarism, AMI, high serum concentration of triglycerides, homocysteine, or C-reactive protein models. In conclusion, our results indicate that young Mexican women with VDD show a higher prevalence of cardiovascular risk factors.

## 1. Introduction

Receptors for vitamin D (VD) and VD activity have been found in many body tissues, suggesting non-calcemic effects of VD related to the regulation of cell proliferation and differentiation, immune response, insulin production, and insulin sensitivity [1,2,3]. These recognized actions of VD suggest that VD plays a role in the prevention of many chronic diseases such as cancer and cardiovascular disease (CVD) [4], but observational studies have not consistently found any association between VD deficiency (VDD) and chronic conditions such as type 2 diabetes mellitus (T2DM), metabolic syndrome, high blood pressure (HBP), and other cardiovascular risk factors [5,6,7,8,9]. Moreover, several supplementation experiments aimed at preventing pathologies associated with VDD have been carried out, but they have yielded contradictory results as regards to the association of VDD with obesity, insulin sensitivity, and insulin secretion, casting doubt on the association of these conditions with VDD [10,11,12]. 

In 2013, the three leading causes of disability-adjusted life years (DALYs) in Mexico were diabetes, ischemic heart disease, and chronic kidney disease, with fasting plasma glucose, high body-mass index (BMI), and HBP being the main risk factors [13]. It is crucial to understand the role of VDD in the occurrence of cardiovascular disease, particularly among young women, in order to help prevent negative health consequences not only for these women in the form of osteoporosis, preeclampsia, or chronic diseases [14,15], but also for their infants, whose bone health and neurological development are influenced by the VD status of their mothers [16,17]. 

Although little research has been performed concerning VDD in women of reproductive age [18], it is well known that long lactation periods and the use of sunscreen constitute risk factors for middle-aged and elderly women [19,20]. Studies in Mexico have revealed high rates of VDD among children and women of reproductive age (≈36%), with an even higher prevalence observed in the presence of obesity, in urban areas and in subjects with low dietary intake of VD [21,22]. 

A recent cohort study in Mexico documented an inverse association between VD intake and cardiovascular risk factors among adults; however, the sample was not nationally representative and 25-OH-D levels were not measured [23]. In addition, most international reports on the relationship between VDD and cardiovascular risks have not addressed several cardiovascular risks together, and few research efforts have focused on young and/or non-pregnant women [6]. Therefore, the objective of this study, conducted among a probabilistic national sample of Mexican women of reproductive age (20 to 49 years old), was to explore whether VDD was associated with sedentarism, overweight, obesity, T2DM, insulin resistance (IR), HBP, high total cholesterol (TC), low high-density lipoprotein cholesterol (HDL-C), high triglycerides (TGs), high homocysteine (Hcy) or C-reactive protein (CRP), and acute myocardial infarction (AMI).

## 2. Materials and Methods

Study population: This analysis was performed among a sample of 3260 women participating in the 2012 National Health and Nutrition Survey (ENSANUT 2012). The ENSANUT 2012 was a probabilistic population-based survey stratified by cluster and representative at the national, regional and urban/rural levels [24]. A detailed description of its sampling method has been published previously [25]. We analyzed the VD data of 3260 women aged 20–49 years with complete information on T2DM, serum concentrations of TC, HDL-C, TG, CRP, IR, and high levels of Hcy. 

### 2.1. Sociodemographic Information

Sociodemographic information was collected using validated questionnaires. A Well-Being Index (WBI) was constructed according to the characteristics of and the property owned by households in a principal component analysis. The first component, representing 40% of total variability with a lambda of 3.4, was divided into tertiles in order to classify the WBI as low, medium, and high [26]. Localities with fewer than 2500 inhabitants were defined as rural and otherwise as urban. As in previous ENSANUTs, the country was divided into three regions: north, center (including Mexico City), and south. An individual was defined as indigenous where one of the members of the household spoke an indigenous language as his/her mother tongue.

### 2.2. Anthropometry

Body weight was measured using an electronic scale with a precision of 100 g, (Tanita Co., Tokyo, Japan), and height using a stadiometer with a precision of 1 mm (Dynatop, Mexico City). These measurements were performed by specialized personnel utilizing the Lohman method [27], and were standardized according to the Habitch method [28].

### 2.3. Blood Samples 

Eight-hour-fasting blood samples were drawn from the winter of 2011 to the spring of 2012, between the latitudes of 14° 54’ and 32° 31’ N. Blood samples were drawn from an antecubital vein and collected in evacuated tubes. Serum was separated by in situ spinning-down at 3000 g. Serum samples were immediately stored in codified cryovials and preserved in liquid nitrogen until delivery to the Central Nutrition Laboratory at the National Institute of Public Health (INSP), in Cuernavaca, Morelos, Mexico. Thereafter, the samples were preserved at −70°C in a deep freezer until chemical analysis. 

A chemiluminescence microparticle immunoassay was used to measure serum 25-OH-D, with intra- and inter-assay coefficient-of-variation (CV) results of 1.34 and 3.69%, respectively. This method has proved acceptable compared to LC/MS/MS (*r* = 0.73) [29]. Quality control was performed according to the reference standard serum NIST 968E of the National Institute of Standards & Technology. The serum concentrations of TC were measured using an enzymatic and oxidation method, glucose by the glucose oxidase technique, HDL-C by direct enzymatic assay after eliminating chylomicrons from the sample, TG by lipase hydrolysis, and Hcy and CRP using ultrasensitive monoclonal antibodies in an Architect CI8200 automatic analyzer (Abbott Lab, Michigan, MI, USA). The intra- and inter-assay CV results were 1.05 and 1.97% for glucose, 2.2% and 5.7% for TC, 3.5% and 5.02% for TG, 5.3 and 7.4% for HDL-C, 3.7% and 4.6% for Hcy, and 0% and 1.2% for CRP, respectively. For insulin, the intra-assay CV was 1.28 uU/mL.

### 2.4. Vitamin D Deficiency

VDD was defined as a serum level of 25-OH-D <50 nmol/L (<20 ng/mL), as in most studies in Latin America [3,30]. 

### 2.5. Definitions of Outcome Variables

T2DM was defined as diabetes previously diagnosed by a physician or a fasting blood glucose level of ≥126 mg/dL [31]. 

Serum biomarkers. The following were considered abnormal serum concentrations: TC >200 mg/dL, TG >150 mg/dL, and HDL-C ≤50 mg/dL [32]. Hcy >10.4 nmol/L was considered abnormal [33]. Inflammation was determined where CRP was >5 mg/dL [34], and IR was defined when homeostatic model assessment of IR (HOMA-IR) was ≥3.8 [35].

Overweight and obesity were based on the body mass index (BMI) as classified by the WHO guidelines (normal BMI ≤24.9 kg/m^2^; overweight 25–29.9 kg/m^2^, and obesity >30 kg/m^2^) [36].

HBP was defined as a previous diagnosis by a physician of hypertension or a systolic blood pressure >140 mmHg and/or a diastolic blood pressure >90 mmHg [37].

AMI was defined when acute myocardial infarction was self-reported by the subject. 

Sedentarism. A lifestyle was defined as sedentary where a woman was classified as having low physical activity according to a validated International Physical Activity Questionnaires (IPAQ) [38]. 

### 2.6. Statistical Analysis

VDD prevalence rates were reported as proportions with a confidence interval (CI) of 95%; risks were expressed as odds ratios (ORs) with a CI of 95%. The significance level was established at an alpha <0.05, and regression models were adjusted by age, BMI, dwelling (urban/rural), geographical region, ethnicity, WBI, and sedentarism, as well as by the survey design, using the module SVY of STATA SE v14 (College Station, TX, USA, 2013).

Ethical aspects. The ENSANUT 2012 protocol was reviewed and approved by the Research, Ethics and Biosecurity Committees of the National Institute of Public Health in Mexico. Prior informed consent letters were signed by all participants.

## 3. Results

### 3.1. Characteristics of the Sample

Serum 25-OH-D levels were estimated for 3260 women representing 19 million Mexican women, aged 20–49 years. In this subsample, 33.4% suffered from overweight, 36.4% from obesity and 25% were sedentary. The prevalence rates for T2DM and HPB were 7.6% and 19.5%, respectively. The overall prevalence of cardiovascular risk factors was 48.3%, and within this group, 80.9% had a low HDL-C, 37.1% high TC, 37.1% high TG, 11.8% high Hcy, and 21.6% high CRP (Table 1).

### 3.2. Prevalence of VDD

The overall prevalence of VDD was 37.2%, significantly higher among women with T2DM compared to women without it (58.6% vs. 34.5%, *p* < 0.05). VDD was also higher among women with obesity (42.5%) vs. normal BMI (29.9%, *p* < 0.05), and with high TC (45.6 vs. 33.9%, *p* < 0.05) and high TG (41.7 vs. 33.8%, *p* < 0.09). No differences emerged regarding the relationships between VDD prevalence and the other cardiovascular risk factors evaluated independently (Figure 1).

### 3.3. Cardiovascular Risk Factors and VDD

We analyzed adjusted logistic models introducing each chronic non-communicable disease and cardiovascular risk factor as dependent variables together with VDD as the independent variable (Figure 2 and Appendix A). We found that the risk of obesity was significantly higher in women with VDD (OR: 1.53, *p* < 0.05) than in women without. The risk was also higher when IR (OR: 1.48, *p* < 0.05), T2DM (OR: 2.58, *p* < 0.05) or high TC (OR: 1.45, *p* < 0.05) was present. These models were adjusted by other cardiovascular risk factors as confounding variables. No association was significant between VDD and the rest of the cardiovascular risk factors. 

In estimating the prevalence of severe VDD (<20 nmol/L), we found that it was notably low (2.3%, 95% CI 1.3–3.9) and had no significant relationship with cardiovascular outcomes (data not shown). We also tested interactions between VDD and age, BMI and sedentarism, and stratified analysis by oral use of hypoglycemic (*n* = 154), antihypertensive (*n* = 129), and hypolipemic drugs (*n* = 210), as well as by menstrual cycle status, and no impact was observed; therefore, the results were not presented.

## 4. Discussion

Our study found that VDD was associated with a higher risk for T2DM, obesity, and high TC concentrations in young Mexican women. The major finding of our analysis was the strong association between VDD and a greater risk of T2DM, a result also yielded by other cross-sectional studies [39,40,41,42]. On the other hand, some published clinical trials have reported an association between VDD and increased insulin resistance, metabolic syndrome hyperglycemia [43,44,45], while others have not [12,46].

Several researchers have found an association between intake of VD and calcium and the prevention of T2DM [11]. Other researchers have administered cholecalciferol weekly for six months and found no difference in insulin response or insulin sensitivity in adults at risk of diabetes mellitus [12]. However, for those with prediabetes, VD supplementation has been shown to improve insulin sensitivity [12,47]. Such an effect could be the result of the interaction of calcium fluxes with the VD receptors in the β-cells, needed for the optimal secretion of insulin [48,49]. 25-OH-D induces synchronous Ca^2+^ oscillations with a pattern of pulsatile insulin secretion from the β-cells [50,51]. This may explain the lower level of insulin sensitivity and secretion, as well as all levels of glucose intolerance. Our study showed that IR was significantly associated with VDD and T2DM. 

An association between VDD and increased adiposity among Mexican women has previously been demonstrated [22]. Our analysis revealed that VDD was 12.6 percentage points higher in obese women than in those with a normal BMI. We hypothesized that such an association could increase the risk for high rates of TSDM, metabolic syndrome, hypertension, and dyslipidemias found in the Mexican population [52,53]. However, it is not yet known if VDD is a cause or a consequence of obesity in humans, and an ample review of the literature provides evidence for both arguments [54].

Although subjects with a higher BMI had greater skin exposure for VD conversion, an experimental exposure to UV rays produced half the amount of serum VD in women with a high as opposed to a normal BMI [55]. Thus, it is possible that the amount of VD previously contained in the body fat of obese women was large enough to interfere with the conversion or incorporation of 25-OH-D into the serum pool. In a clinical trial involving calcium and VD supplements among overweight and obese individuals, was observed a significant reduction of visceral fat [56]. Obese subjects have greater adipose stores of VD. This enlarged adipose mass in obese individuals serves as a VD reservoir, and the increased amount of VD required to saturate this depot may predispose obese individuals to have an inadequate level of 25-OH-D [57]. 

Our study found that women with VDD had a 1.43 times greater risk (95% CI 1.03–1.99) for high levels of TC compared to women with sufficient VD concentrations (*p* = 0.04). This may be a result of the suppressive effect of VD on the parathyroid hormone (PTH), which reduces lipolysis. Similarly, it has been suggested that VD increases calcium concentrations, reducing the synthesis de novo and releasing hepatic TG [58]. Observational studies have reported that women with VDD have a higher risk of lipid concentration as compared to women with adequate VD status [59,60,61]. However, in a meta-analysis involving 12 clinical trials with 1346 subjects, VD supplementation was associated with improved LDL-C concentrations, but not with improved TG, HDL-C or TC levels [62]. Meanwhile, a recent randomized controlled trial in postmenopausal women showed positive effects in VD status, bone mineral density, glucose, TC, LDL-C, and apolipoprotein B100 after 24 months of intake of vitamin D-enriched milk [63].

Our study found no association between HBP and VDD, reflecting the findings of other studies [64,65]. However, some observational and randomized clinical trials have found a negative correlation between the two variables [66,67]. 

The cross-sectional nature of the 2012 ENSANUT makes it difficult to establish causality. However, our results are representative for the entire population of Mexican women aged 20–49 years and for countries with similar socioeconomic characteristics. Another limitation was that data for some variables were collected by means of self-reporting, potentially introducing a degree of measurement bias. Nonetheless, data for all variables were collected in a similar manner; thus, the bias will not affect results for the target population, and the results might even underestimate a potential association. In our analyses, we adjusted results by menstrual cycle status with no significant alteration in the associations, but we were unable to adjust by the current use of estrogen-containing contraception that has been associated with increases in 25-OH-D levels in other populations [19,68]. Nevertheless, in our sample 88% of women had been pregnant and 28% had been sterilized after childbirth. According to the National Survey Report, in 2012 the rate of use of hormonal contraceptives among Mexican women was 12.5% for those aged 20–29 years old, 7.3% among 30–39 year-olds, and 4.6 among 35–49 year-olds, while 42% to 53% reported not using any contraceptive method [69]. Among the strengths of this study is its population-based sample, that allows for representativeness for Mexican women 20–49 years, and the fact that the study includes information on VD serum levels and CVD risk factors.

## 5. Conclusions

The prevalence of VDD among women 20 to 49 years old is a public health problem in Mexico. Obesity, T2DM, IR, and high TC were found to be associated with VDD. Further research is necessary to assess the biological mechanisms underlying all of these factors and their association with VDD. Also, this study provides scientific evidence supporting the need for public health and nutrition interventions aimed at improving the vitamin D status of Mexican women.

## Figures and Tables

**Figure 1 nutrients-11-01211-f001:**
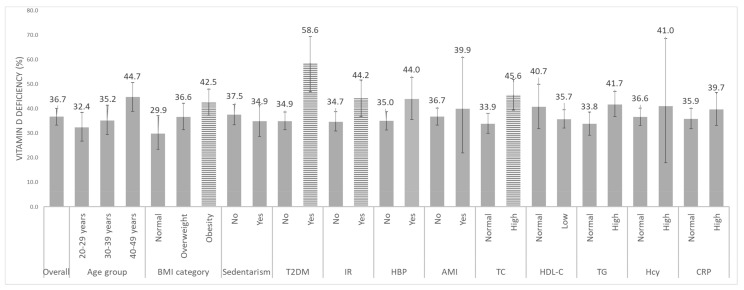
Adjusted prevalence (and 95% confidence interval) of vitamin D deficiency by age group and cardiovascular risk factor among Mexican women aged 20–49 years. BMI: body mass index; T2DM: type 2 diabetes mellitus; IR: insulin resistance; HBP: high blood pressure; TC: total cholesterol; HDL-C: high density lipoprotein; TG: triglycerides; AMI: acute myocardial infarction; Hcy: homocysteine; CRP: C-reactive protein. Horizontal bars indicate a statistical difference (*p* value < 0.05) among vitamin-D-deficiency prevalence rates in the subcategories. Reference subcategories are 20–29 years for age group; “Normal” for BMI, TC, HDL-C, TG, Hcy, and CRP; and “No” for sedentarism, T2DM, IR, HBP, and AMI. Logistic regression model was adjusted by area (urban/rurality), region of the country, Wellbeing index tertiles, ethnicity and all cardiovascular risk factors.

**Figure 2 nutrients-11-01211-f002:**
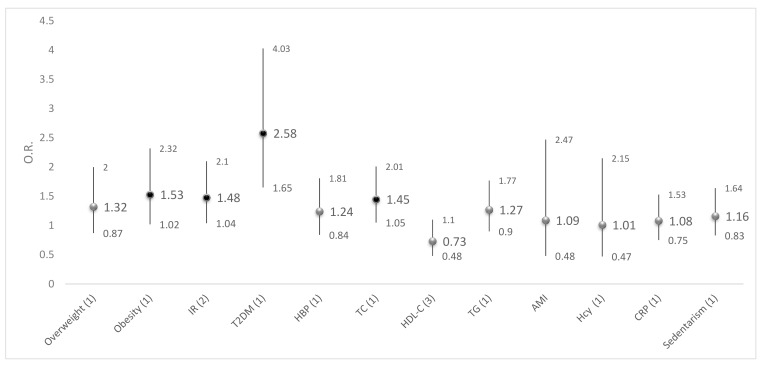
Odds ratios (OR) and 95% confidence interval adjusted for cardiovascular risk factors by vitamin D deficiency in Mexican women aged 20–49 years. BMI: body mass index; PA: physical activity; T2DM: type 2 diabetes mellitus; IR: insulin resistance; HBP: high blood pressure; TC: total cholesterol; HDL-C: high-density lipoprotein; TG: triglycerides; AMI: acute myocardial infarction; Hcy: homocysteine; CRP: C-reactive protein. (1) All models were adjusted by area (urban/rurality), region of the country, Wellbeing index tertiles, ethnicity and the following risk factor variables: HDL-C (<50 mg/dL), TG (<150 mg/dL), TC (<200 mg/dL), HBP, T2DM, CRP (<5 mg/dL), Hcy (<10 umol/L), sedentarism, overweight–obesity, IR, and AMI, except by the risk factor studied in the particular model. (2) Adjusted by all variables in model 1 excluding T2DM. (3) Adjusted by all variables in model 1 excluding TC. Black circle indicate a statistical difference (*p* value < 0.05)

**Table 1 nutrients-11-01211-t001:** Characteristics and distribution of the sample.

Variable	Subgroup	% (95% CI) *
Age (years)	20–29	36.2 (33, 39.6)
	30–39	37.1 (33.8, 40.5)
	40–49	26.8 (24.2, 29.6)
Dwelling	Rural	21.5 (19.7, 23.4)
	Urban	78.6 (76.7, 80.4)
Region of the country	North	22.4 (20.7, 24.2)
	Center	47.7 (45.2, 50.2)
	South	30.1 (27.9, 32.3)
Well-Being Index	Tertile 1 (lower)	25.5 (23.2, 27.9)
	Tertile 2	31.6 (28.8, 34.6)
	Tertile 3 (higher)	43.1 (39.7, 46.6)
Ethnicity	No	94.7 (93.4, 95.7)
	Yes	5.3 (4.3, 6.6)
BMI category	Normal	30.3 (27.2, 33.5)
	Overweight	33.4 (30.4, 36.6)
	Obesity	36.4 (33.4, 39.6)
Sedentarism	No	70.4 (67.2, 73.3)
	Yes	29.6 (26.7, 32.8)
T2DM	No	92.4 (90.4, 94.0)
	Yes	7.6 (6.0, 9.7)
HBP	No	87.9 (85.3, 90.0)
	Yes	19.5 (17.0, 22.3)
TC	<200 mg/dL	75.8 (72.8, 78.6)
	≥200 mg/dL	24.2 (21.5, 27.2)
HDL-C	≥50 mg/dL	19.1 (16.3, 22.2)
	<50 mg/dL	80.9 (77.7, 83.7)
TG	<150 mg/dL	62.9 (59.8, 66.0)
	≥150 mg/dL	37.1 (34.0, 40.3)
AMI	No	98.7 (98.2, 99.1)
	Yes	1.20 (0.81, 1.76)
Hcy	<10.4 nmol/L	87.9 (85.3, 90.0)
	≥10.4 nmol/L	12.1 (9.9, 14.6)
IR	<3.8 HOMA-IR	79.4 (76.2, 82.3)
	≥3.8 HOMA-IR	20.6 (17.7, 23.8)
CRP	<5 g/L	78.4 (75.8, 80.9)
	≥ 5 g/L	21.6 (19.1, 24.2)

*N* sample = 3260, *n* expanded = 19,336,909. CI: Confidence Interval; BMI: body mass index; T2DM: type 2 diabetes mellitus; HBP: high blood pressure; TC: total cholesterol; HDL-C: high-density lipoprotein; TG: triglycerides; AMI: acute myocardial infarction; Hcy: homocysteine; IR: insulin resistance; CRP: C-reactive protein * Expanded % and 95% CI.

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
