# Peer review of "Cardiovascular Risk Factors and Their Association with Vitamin D Deficiency in Mexican Women of Reproductive Age"

_nutrients, 2019, doi:10.3390/nu11061211_

Reviewer 1 Report

This manuscript aims at associating vitamin D deficiency and cardiovascular risk factors in Mexican women at ages of reproduction. The study of the relationship between vitamin D deficiency and cardiovascular disease is a topic of interest and of relative novelty.

Major comments:

Authors aim at associating VDD with cardiovascular risk factors. However, some analyses as reflected in the discussion seem to invert this association (i.e. lines 154-156: “We report in this manuscript women with T2-Diabetes, obesity, high concentrations of T-chol and TG were positively associated with higher risk for VDD. However, women with hypertension, high Hcy and high CRP had no significant risks for VDD.”). Authors should clearly state the direction of the association and perform the statistical analyses and discuss the results in accordance with the hypothesis.

Sample size is confusing. In Methods lines 72-74, authors state that “8 h fasting blood samples were drawn between the winter of 2011 and the spring of 2012, and between the latitudes 14°54’ and 32°31’ N., in 30% of the women 20-40 y of ENSANUT  2012 (n=4,000)”. This means that vitamin D has been quantified in 1200 samples.  However, in Results lines 119-12, is it explained that “Serum 25-OH-D were determined in 3,260 women, representing 19 million Mexican women, 20-49 years old.”. Please clarify this important issue. In addition, authors should rewrite this last sentence, because this subsample accounts for only a 0.02% of this population.

Methods line 90: VDD was defined as serum 25-OH-D < 50 nmol/L (<20 ng/mL). Some authors consider vitamin D deficiency as <10 ng/mL. Why authors did choose this cut-off? Perhaps authors could report a supplemental analysis with 10 ng/mL as a cut-off for VDD.

Table 1. Data presentation is confusing. What is statistics comparing? Clustered varibles among tertiles? It is unclear. Moreover, a descriptive Table should depict n(%) and not %(95%). Please revise it. What is the purpose of providing this information: “n expanded n=19,336,909”?

Figure 1. Please expand abbreviations used at the end of the footnote.

Results, lines 140-149. The models and multi-variable adjustments in each model are unclear. Please provide a table with each model, specifying covariates, and the corresponding OR.

Line 212: Delete “probabilistic design”.

Minor comments:

Line 29, replace “end” by “and”.

Line 55: Replace “con” by “on”.

Line 112: Replace “controlled” by “adjusted”.

Lines 112-114: The main verb of the sentence is missing.

Author Response

Comments and Suggestions for Authors

This manuscript aims at associating vitamin D deficiency and cardiovascular risk factors in Mexican women at ages of reproduction. The study of the relationship between vitamin D deficiency and cardiovascular disease is a topic of interest and of relative novelty.

Major comments:

Authors aim at associating VDD with cardiovascular risk factors. However, some analyses as reflected in the discussion seem to invert this association (i.e. lines 154-156: “We report in this manuscript women with T2-Diabetes, obesity, high concentrations of T-chol and TG were positively associated with a higher risk for VDD. However, women with hypertension, high Hcy and high CRP had no significant risks for VDD.”). Authors should clearly state the direction of the association and perform the statistical analyses and discuss the results in accordance with the hypothesis.

Thank you for the observation. We have modified the text in the correct direction according to the adjusted models.

“We report in this manuscript VDD was associated with a higher risk for T2DM, obesity and high concentrations of TC” 

Sample size is confusing. In Methods lines 72-74, authors state that “8 h fasting blood samples were drawn between the winter of 2011 and the spring of 2012, and between the latitudes 14°54’ and 32°31’ N., in 30% of the women 20-40 y of ENSANUT  2012 (n=4,000)”. This means that vitamin D has been quantified in 1200 samples. 

We have corrected the paragraph, our sub-sample were 3,260 women which is a 30% of the complete sample of the national survey.

“Study population: This analysis was carried out in a sample of 4,000 women participating in the National Health and Nutrition Survey-2012 (ENSANUT 2012). ENSANUT 2012 is a probabilistic population-based survey stratified by cluster, and representative at the national, regional and urban/rural levels.[24]”

However, in Results lines 119-12, is it explained that “Serum 25-OH-D was determined in 3,260 women, representing 19 million Mexican women, 20-49 years old.”. Please clarify this important issue. In addition, authors should rewrite this last sentence, because this subsample accounts for only a 0.02% of this population.

ENSANUT sample has a sub-sample with serum and diet information collected. In our study, we used the information of the sub-sample that was 30% of the whole ENSANUT participants. But, to avoid confusion, we eliminated the percentage information of the subsample because provides as well the whole sample, national representativeness. The methodology of the sample calculation and survey design of ENSANUT was published elsewhere. Briefly, probabilistic surveys with national representativeness (as its ENSANUT) allows to expand the results of the sample to the overall population. In our case, 3,260 participants (30% of the whole sample of women in ENSANUT) represented a population of 19 million of Mexican women. 

Methods line 90: VDD was defined as serum 25-OH-D < 50 nmol/L (<20 ng/mL). Some authors consider vitamin D deficiency as <10 ng/mL. Why authors did choose this cut-off? Perhaps authors could report a supplemental analysis with 10 ng/mL as a cut-off for VDD.

In order to be comparable with other studies we used<50 nmol/L cut off point because it has been the most used cut off point to define Vitamin D deficiency in Latinamerican studies, In the case of 10 ng/mL, is commonly used for severe deficiency and to evaluate the risk of bone outcomes. Nevertheless, to respond to your interesting point we have included in the results section the overall prevalence of severe vitamin D deficiency and included a reference of studies about VDD prevalence in Latin America.

Line 90: “Vitamin D deficiency: VDD was defined as serum 25-OH-D < 50 nmol/L (<20 ng/mL) as in most of the studies carried out in Latin America. [3, 30]”

Line 179:  We calculated the prevalence of severe VDD (<20 nmol/L) and we found that it was very low (2.3%, 95%CI 1.3-3.9) with no significant relation with cardiovascular outcomes (data not shown).

Table 1. Data presentation is confusing. What is statistics comparing? Clustered varibles among tertiles? It is unclear.

The aim of Table 1 was to present characteristics of the sample in order to describe the distribution of their strata and show that is information similar to the Mexican Census. Also, it aimed to provide information that has not been published previously as are the prevalence of cardiovascular risk factors.

Moreover, a descriptive Table should depict n(%) and not %(95%). Please revise it.

Thank you for the comment. It is common to report % and 95% confidence intervals (CI) to describe complex samples as it is our survey information. Several articles of ENSANUT have been reported as we did in many journals. We hope we convinced you in order to maintain Table 1 as originally depicted. All the prevalences are estimated from the overall N sample (3,260) that is depicted in the title of table 1. We also think that 95% CI shows better precision at the estimations of the population (19 million).

What is the purpose of providing this information: “n expanded n=19,336,909”?

We provided n expanded information because is the amount of Mexican woman that our sample allows us to make an inference to according to the sampling design of the survey. For example, we are 95% sure that the prevalence of T2DM in 19 million young Mexican women was between 6% and 9.7% in 2012. For this reason, our study is relevant as evidence for public health interventions related to improve vitamin D status in Mexican women.

Figure 1. Please expand abbreviations used at the end of the footnote.

Thank you for the observation, we have included a footnote in Figure 1 with the expand abbreviations:

BMI: Body Mass Index, T2DM; Type 2 Diabetes Mellitus, IR; insulin resistance, HBP; high blood pressure, TC; total cholesterol, HDL-C; High density lipoprotein, TG; triglycerides, AMI; acute myocardial infarction, Hcy; homocysteine, CRP; C-reactive protein.

Results, lines 140-149. The models and multi-variable adjustments in each model are unclear. Please provide a table with each model, specifying covariates, and the corresponding OR.

In order to clarify the results, we have added according to your suggestion, a supplementary table that shows the complete model for each cardiovascular outcome. (Supplementary Table 1).

Line 212: Delete “probabilistic design”.

We modified text as suggested

Minor comments:

Line 29, replace “end” by “and”.

We modified text as suggested

Line 55: Replace “con” by “on”.

We modified text as suggested

Line 112: Replace “controlled” by “adjusted”.

We modified text as suggested

Lines 112-114: The main verb of the sentence is missing.

We have included the verb missing.

“Regression models were controlled adjusted by age, BMI, dwelling (urban/rural),…”

Reviewer 2 Report

Epidemiological study to evaluate the influence of vitamin D deficiency on cardiovascular risk factors in a population of young Mexican women (20-49 y).

The main finding is that vitamin D deficiency (25OHD<50 nmol / L) was significantly more frequent in women with obesity and type 2 diabetes (T2DM). Although the population sample is large, the results provide information that is already know and published. 

The writing of the manuscript should be improved and there are errors that must be corrected

Abstract

The p values and confidence intervals of the results shown should be indicated.

The abbreviation for diabetes mellitus type 2 should be "T2DM" and for total cholesterol "TC" throughout the manuscript.

Triglyceride results should be eliminated from the abstract because they do not reach statistical significance.

The conclusion of the abstract should be something like "Our results indicate that women with VDD have a higher prevalence of cardiovascular risk factors.

Introduction

The introduction of the study should make more emphasis on the consecuences of the VDD in young women, which is the main singularity of the work

Materials and Methods

The definition of diabetes is fasting blood glucose ≥ 126 mg / dl

"Insulin resistance was defined when HOMA was greater that 5". Where does that statement come from?

No information is provided on the pharmacological treatment of the participants

Results

How is the percentage of women with a myocardial infarction explained?

Figure 1 should be revised. HBP values are incomplete

Discussion

In the discussion it is pointed out that TG levels are associated with the VDD but the results do not indicate it. 

Paragraph "In an ample review of the literature, there is insufficient evidence to establish whether: a) VDD is associated with a lean or obese phenotype, b) VDD is a consequence of obesity, or c) the effects of vitamin D on fat tissue are due to interactions with calcium. " is inaccurate and imprecise and should be rewritten. 

Overall, the entire discussion should be rewritten by describing in detail the known relationship of VDD with obesity, dysglicemia and lipid alterations.

Author Response

Comments and Suggestions for Authors

Epidemiological study to evaluate the influence of vitamin D deficiency on cardiovascular risk factors in a population of young Mexican women (20-49 y).

The main finding is that vitamin D deficiency (25OHD<50 nmol / L) was significantly more frequent in women with obesity and type 2 diabetes (T2DM). Although the population sample is large, the results provide information that is already know and published. 

Thank you for your valuable comments. We support the importance of our study as most of the international reports on VDD and cardiovascular risks factors have not been consistently addressed cardiovascular risk simultaneously, which allow avoiding confusion by BMI, age or other common risk factors of cardiovascular diseases and VDD. Also, to our knowledge few of the studies have been done in national representative samples of young non-pregnant women. 

The writing of the manuscript should be improved and there are errors that must be corrected

Thank you for your comment. We have rewrite and corrected suggested sections of the manuscript.

Abstract

The p values and confidence intervals of the results shown should be indicated.

We included p values and 95%CI in the abstract.

“The prevalence of VDD was significantly higher in obese women (42.5%, 95%CI; 37.3-47.9) compared to women with normal BMI (29.9%, 95%CI; 23.5-37.1), in those with high TC (45.6% compared to normal TC (33.9%), and also higher prevalence of VDD in women with Insulin Resistance (44%, 95%CI; 36.9-51.7) or T2DM (58.6%, 95% CI 46.9-69.4) compared to those with normal glycaemia.(?35%, p<0.05) According to individual model to estimate cardiovascular risk by VDD, we found that odds of obesity (OR: 1.38, 95%CI 1.02-2.32), high TC (O.R.:1.43, 95%CI 1.05-2.01), T2DM (O.R.: 2.64, 95%CI 1.65-4.03) or IR (OR: 1.48, 95%CI 1.04-2.10) were significantly higher in women with VDD (p<0.05).”< span="">

The abbreviation for diabetes mellitus type 2 should be "T2DM" and for total cholesterol "TC" throughout the manuscript.

Thank you for the comment. We have homologated through all the manuscript the terms T2DM, TC and also HDL-C, CRP, TG and Hcy.

Triglyceride results should be eliminated from the abstract because they do not reach statistical significance.

Thank you for this observation. In previous analyses we found significance for high triglycerides before adjust by all the covariates included in the final manuscript, now we have eliminated of the abstract the triglyceride results and also we have modified the title to “Cardiovascular risk factors and its association with vitamin D deficiency in Mexican women at reproductive age.”

The conclusion of the abstract should be something like "Our results indicate that women with VDD have a higher prevalence of cardiovascular risk factors.

Thank you, we have modified the conclusion similarly to your suggestion.

“In conclusion, our results indicate that young Mexican women with VDD have a higher prevalence of cardiovascular risk factors.”

Introduction

The introduction of the study should make more emphasis on the consequences of the VDD in young women, which is the main singularity of the work

We have rewritten and re-organized the introduction

Materials and Methods

The definition of diabetes is fasting blood glucose ≥ 126 mg / dl

Thank you for the correction.

"Insulin resistance was defined when HOMA was greater that 5". Where does that statement come from?

Cut off point was incorrect in the text, the analyses were carried out using the cut off ≥3.8, reference was modified.

“IR was defined when HOMA-IR was greater than >=3.85 [35]”

Reference 35. 34.35.        Ascaso JF, Romero P, Real JT, Priego A, Valdecabres C, Carmena R. Insulin resistance quantification by fasting insulin plasma values and HOMA index in a non-diabetic population. Med Clin (Barc). 2001 Nov 3;117(14):530-3.

No information is provided on the pharmacological treatment of the participants

We moved from Statistical analysis to Results the paragraph “We tested several models adjusting for the use of oral hypoglycemic or hypolipemic drugs, menstrual cycle status and CRP>5 mg/L with no significant impact on the regression model estimations, thus they were not presented.”

Results

How is the percentage of women with a myocardial infarction explained?

As it’s a sample of young women but with highly risk for AMI, we found a very low prevalence of AMI.

Figure 1 should be revised. HBP values are incomplete

Thank you for the observation. We included complete values and footnotes in Figure 1

Discussion

In the discussion, it is pointed out that TG levels are associated with the VDD but the results do not indicate it. 

We are sorry for the mistake. We have eliminated results on TG levels found in previous analyses.

Paragraph "In an ample review of the literature, there is insufficient evidence to establish whether: a) VDD is associated with a lean or obese phenotype, b) VDD is a consequence of obesity, or c) the effects of vitamin D on fat tissue are due to interactions with calcium. " is inaccurate and imprecise and should be rewritten. 

We have rewritten many of the discussion paragraphs in order to make it more clear and accurate.

Overall, the entire discussion should be rewritten by describing in detail the known relationship of VDD with obesity, dysglicemia and lipid alterations

Thank you. We included more information in the discussion section following your suggestions.

Round  2

Reviewer 1 Report

The manuscript has partially improved but there are still major concerns to be amended. Overall, the manuscript contains several typos and grammatical errors, please revise it thoroughly. Also please correct at the title: “their associations” (not “its associations”).

Major comments:

If VD is analysed on 3260 women, then why authors state that “This analysis was carried out in a sample of 4,000 women participating in the National Health and Nutrition Survey-2012 (ENSANUT 2012)”? Authors should state that “This analysis was carried out in a sample of 3,260 women participating in the National Health and Nutrition Survey-2012 (ENSANUT 2012)”.

The statistics of Table 1 and Figure 1 still remains unclear. Superscript “a” in Table 1 is comparing 1 characteristic (for instance, Centre region of the country) against women with all of these characteristics: “20-29 yo, Rural, North, Tertile 1 of wellbeing index, no indian ancestry, normal BMI, Sedentary, No DM, No hypertension, TC<200 hdl-cholesterol="">=50 mg/dl, 238 Triglycerides”? For what purpose? Or is comparing the prevalence of categories within one characteristic? Please clarify the statistical analysis in Table 1 footnote. It would make more sense to compare prevalences of their subsample (n=3260) with prevalences of the Total survey or the Mexican Census to clearly state that their subsample is representative of the Total survey and “that allows for national representativeness of 19 million Mexican women 20-49y”, and also to give sense at showing n expanded. Authors should indicate somewhere what n expanded accounts for. Is the 95%CI from the n expanded? If so, it should be mentioned.

Also please clarify what the p value accounts for in Figure 1.

With the new analysis with menstrual cycle, authors should also consider the use of birth control pills or other contraceptive drugs.

Supplemental Table 1 does not show multivariable-adjusted models or “several adjusted logistic models introducing each chronic non-transmissible disease and the cardiovascular risk factors as dependent variable and VDD as independent variable”. This is shown only in the two first rows as unadjusted bivariate model. Please remake Supplemental Table 1 according to the Results section. Authors should explain what denote superscripts “a” and “b”. What does SES mean? Tertile 1 of SES is the higher or the lower SES? Please specify.

Minor comments:

 Abstract: please include the three items: 1) %, 2)95%CI and 3) p value for each cardiovascular risk factor.

Please expand DALY (it is mentioned only once).

Author Response

The manuscript has partially improved but there are still major concerns to be amended. Overall, the manuscript contains several typos and grammatical errors, please revise it thoroughly. Also please correct at the title: “their associations” (not “its associations”).

Manuscript was reviewed and corrected by a native speaker English.

Major comments:

If VD is analysed on 3260 women, then why authors state that “This analysis was carried out in a sample of 4,000 women participating in the National Health and Nutrition Survey-2012 (ENSANUT 2012)”? Authors should state that “This analysis was carried out in a sample of 3,260 women participating in the National Health and Nutrition Survey-2012 (ENSANUT 2012)”.

Thank you for your comment, we included the total sample of women with 25-OH-D data available (n=4000), but certainly, the analyses were carried out in 3,260 women with cardiovascular risk factors and vitamin D data. We have changed 4,000 for 3,260.

The statistics of Table 1 and Figure 1 still remains unclear. Superscript “a” in Table 1 is comparing 1 characteristic (for instance, Centre region of the country) against women with all of these characteristics: “20-29 yo, Rural, North, Tertile 1 of wellbeing index, no indian ancestry, normal BMI, Sedentary, No DM, No hypertension, TC<200 hdl-cholesterol="">=50 mg/dl, 238 Triglycerides”? For what purpose? Or is comparing the prevalence of categories within one characteristic? Please clarify the statistical analysis in Table 1 footnote. It would make more sense to compare prevalences of their subsample (n=3260) with prevalences of the Total survey or the Mexican Census to clearly state that their subsample is representative of the Total survey and “that allows for national representativeness of 19 million Mexican women 20-49y”, and also to give sense at showing n expanded. Authors should indicate somewhere what n expanded accounts for. Is the 95%CI from the n expanded? If so, it should be mentioned.

Thank you for your observations, to avoid confusion we decided to eliminate statistical comparison superscripts and we have included the complementary information (% and 95%CI) for each cardiovascular risk factor, and also we added a Footnote to explain that information in table 1 are expanded estimations as follows:

n expanded=19,336,909

% (95 %C.I.)*

*Expanded % and 95% CI

Also please clarify what the p value accounts for in Figure 1.

We have included the reference subcategory used for each statistical comparison.

Horizontal bars indicates a statistical difference (p value<0.05) among vitamin D deficiency prevalence among subcategories.

Reference subcategories are: for age group; 20.29 y, for BMI, TC, HDL-C, TG, Hcy and CRP; “Normal”, and for Sedentarism, T2DM, IR, HBP and AMI; “No”. Reference subcategories are: for age group; 20.29 y, for BMI, TC, HDL-C, TG, Hcy and CRP; “Normal”, and for Sedentarism, T2DM, IR, HBP and AMI; “No”.

With the new analysis with menstrual cycle, authors should also consider the use of birth control pills or other contraceptive drugs.

Reviewer 2 Report

The revised manuscript has improved many of the observed deficiencies of the original manuscript. The authors highlight the characteristics of the study population, young premenopausal women, which increases the interest of work.

The issues raised have been adequately addressed. However, several typographical errors are identified that must be corrected. In the discussion, to extend the explanation of the findings in the lipid profile it would be advisable to mention two recent articles: Schnatz PF, et al, Lipids Health Dis 2012 and Reyes-Garcia R, et al J Womens Health (Larchmt). 2018

Author Response

The revised manuscript has improved many of the observed deficiencies of the original manuscript. The authors highlight the characteristics of the study population, young premenopausal women, which increases the interest of work.

Thank your for your comments.

The issues raised have been adequately addressed. However, several typographical errors are identified that must be corrected.

Manuscript was reviewed and corrected by a native speaker English.

In the discussion, to extend the explanation of the findings in the lipid profile it would be advisable to mention two recent articles: Schnatz PF, et al, Lipids Health Dis 2012 and Reyes-Garcia R, et al J Womens Health (Larchmt). 2018

Thank you for the suggestions. We have incorporated Reyes-Garcia R results in our discussion.

Lines 262-267: However, in a meta-analysis involving 12 clinical trials with 1,346 subjects, VD supplementation was associated with improved LDL-C concentrations, but not with improved TG, HDL-C or TC levels [62]. Meanwhile, a recent randomized controlled trial in postmenopausal women showed positive effects in VD status, bone mineral density, glucose, TC, LDL-C, and apolipoprotein B100 after 24 months of intake of vitamin D-enriched milk [63].